

# Functional polymorphisms of the APOA1/C3/A4/A5-ZPR1-BUD13 gene cluster are associated with dyslipidemia in a sex-specific pattern

Wei Bai[1], Changgui Kou[1], Lili Zhang[1], Yueyue You[1], Weiying Yu[1], Wanqing Hua[1], Yuanyuan Li[1], Yaqin Yu[1], Tiancheng Zhao[2] and Yanhua Wu[1,3]

[1] Department of Epidemiology and Biostatistics, School of Public Health, Jilin University, Changchun, Jilin Province, China
[2] Department of Endoscopy Center, China-Japan Union Hospital of Jilin University, Changchun, Jilin Province, China
[3] Division of Clinical Research, First Hospital of Jilin University, Changchun, Jilin Province, China

Corresponding authors
Tiancheng Zhao,
zhaotiancheng@jlu.edu.cn
Yanhua Wu, wuyanhua@jlu.edu.cn

## ABSTRACT

**Background:** Dyslipidemia contributes to the risk of many diseases, including stroke, cardiovascular disease and metabolic-related diseases. Previous studies have indicated that single nucleotide polymorphisms (SNPs) are associated with different levels of serum lipid. Therefore, this study explored the relationship between the *APOA1/C3/A4/A5-ZPR1-BUD13* gene cluster gene polymorphisms and dyslipidemia in the total sample population and stratified by genders in a northeast Chinese population.

**Methods:** A total of 3,850 participants from Jilin Province, China, were enrolled in our study, and their serum lipid levels were measured. Six functional SNPs (*APOA1* rs5072, *APOC3* rs5128, *APOA4* rs5104, *APOA5* rs651821, *ZPR1* rs2075294 and *BUD13* rs10488698) were genotyped using polymerase chain reaction and MALDI-TOF-MS. Logistic regression analysis was performed to explore the relationship of *APOA1/C3/A4/A5-ZPR1-BUD13* gene cluster gene polymorphisms with dyslipidemia. Linkage disequilibrium and haplotype analyses were performed with the SNPStats program and Haploview software.

**Results:** All SNPs conformed to Hardy–Weinberg equilibrium. Logistic regression analysis revealed that rs5072, rs5128 and rs651821 were associated with hypertriglyceridemia, rs5104 and rs651821 were associated with low-HDL cholesterolemia in overall group. rs651821 was associated with hypertriglyceridemia and low-HDL cholesterolemia in both the male and female group. However, among females, rs5072 was observed to be associated with hypertriglyceridemia. Haplotype analysis showed that haplotypes TGCCGC and CAGCGC were associated with dyslipidemia in the overall, male and female groups.

**Conclusion:** SNPs in the *APOA1/C3/A4/A5-ZPR1-BUD13* gene cluster were associated with dyslipidemia. Furthermore, the association of *APOA1* rs5072 in this gene cluster with dyslipidemia differed between genders; thus, additional studies are needed to confirm this conclusion, and the mechanisms underlying these results warrant further exploration.

## BACKGROUND

Dyslipidemia, which is characterized by abnormally increased total cholesterol (TC), triglycerides (TG), low-density lipoprotein cholesterol (LDL-C) and decreased high-density lipoprotein cholesterol (HDL-C) levels, is a well-established risk factor for cardiovascular diseases (*El Hajj et al., 2016*; *Parhofer, 2015*; *Rai et al., 2016*) and contributes to metabolic syndrome (MetS), stroke and other metabolism-related diseases (*Albai, Roman & Frandes, 2017*; *Amalakanti et al., 2016*; *Wang et al., 2017*; *Wu et al., 2016*). A nationally representative survey in China indicated that the prevalence of high TC, high LDL-C, low HDL-C and high TG in Chinese adults over the age of 18 was 6.9, 8.1, 20.4 and 13.8%, respectively (*Zhang et al., 2018*). Previous studies involving epidemiological surveys found that diet, exercise and other environmental risk factors may be associated with abnormal serum lipid levels (*Tan et al., 2016*). However, genetic factors could also partially explain the development of dyslipidemia (*Dron & Hegele, 2016*, *2018*; *Weiss et al., 2006*).

The *APOA1/C3/A4/A5-ZPR1-BUD13* (APO, apolipoprotein; ZPR1, zinc finger; BUD13, BUD13 homolog) gene cluster is located on chromosome 11q23.3, and the apolipoprotein gene cluster at this location has been revealed to be more often associated with lipid traits via genome-wide association study (GWAS) (*Braun et al., 2012*; *Willer et al., 2013*). APOA1 synthesis and lipolysis promote HDL-mediated reverse cholesterol transport by generating cholesterol acceptors (*Reichl & Miller, 1989*); therefore, APOA1 plays an important part in lipid metabolism, especially that of HDL cholesterol. The major function of APOC3 in lipid metabolism is to inhibit lipoprotein lipase (LPL); thus, the concentrations of plasma APOC3 are positively associated with TG concentrations (*Ooi et al., 2008*). APOA4 modulates the activation of LPL by APOC2 (*Goldberg et al., 1990*) and participates in reverse cholesterol transport (*Stein et al., 1986*). The transcription of APOA4 is regulated synchronously with that of APOA1, supporting the relationship between these apolipoproteins (*Malmendier, Alaupovic & Brewer, 1991*) and APOA4 is associated with TG and HDL levels. APOA5 is a component of HDL and regulates TG levels via hydrolysis of TG-rich lipoproteins and endocytosis of lipoprotein remnants (*Forte, Shu & Ryan, 2009*). *BUD13* encodes the BUD13 homolog protein, which is one of the subunits of the splicing factor that affects nuclear pre-mRNA retention (*Brooks et al., 2009*). *ZPR1* encodes a regulatory protein involved in cell proliferation and signal transduction (*Galcheva-Gargova et al., 1996*). In summary, based on the above mechanisms, this gene cluster can influence lipid levels, therapy contributing to the risk of dyslipidemia.

Relevant studies have been conducted to explore the relationship between serum lipid levels and genetic loci in this gene cluster among different ethnic groups (*Aung et al., 2014a*, *2014b*; *Hsueh et al., 2017*; *Suárez-Sánchez et al., 2017*; *Yin, Li & Lai, 2011*). However, to the best of our knowledge, there have been few studies conducted in
northeast Chinese populations that explore the relationship between the *APOA1/C3/A4/A5-ZPR1-BUD13* gene cluster and dyslipidemia. Functional single nucleotide polymorphism (SNP) may influence gene and protein expression. This study assessed the association of tag and well-studied SNPs (*APOA1* rs5072, *APOC3* rs5128, *APOA4* rs5104, *APOA5* rs651821, *ZPR1* rs2075294 and *BUD13* rs10488698) selected from the Han Chinese data in Haploview (http://hapmap.ncbi.nlm.nih.gov/) and based on previous studies (*Wu et al., 2015*, *2016*) with dyslipidemia in the overall sample and stratified by genders in a large sample from a Han Chinese population.

## MATERIALS AND METHODS

### Study population

The Project of Present Situation and Change Forecast of Disease Spectrum in Jilin Province, China, was conducted from June 2012 to August 2012 among individuals aged 18–79 years in all nine areas of Jilin Province. This project was a cross-sectional and representative survey of Jilin Province, China, which aimed to evaluate the prevalence and risk factors associated with chronic diseases (*Wang et al., 2015*). The genetic associations results for dyslipidemia-related diseases have been reported in previous studies (*Su et al., 2016*; *You et al., 2017*, *2018*). The whole sample size of the survey was 21,435, and the participants in our present study were randomly chosen from the whole sample. They were unrelated members of the Han population, and individuals with a family history of dyslipidemia and other metabolic diseases or without complete serum lipid data were excluded; the participants were excluded if they were being treated with any drugs that may affect lipid parameters. In total, 3,850 participants (1,927 males and 1,923 females) were enrolled in the association study between the *APOA1/C3/A4/A5-ZPR1-BUD13* gene cluster and dyslipidemia.

Dyslipidemia was assessed according to the Guidelines on Prevention and Treatment of Dyslipidemia in Chinese Adults (*Joint Committee for Developing Chinese guidelines on Prevention and Treatment of Dyslipidemia in Adults, 2007*): hypertriglyceridemia, TG $\geq$2.26 mmol/L; hypercholesterolemia, TC $\geq$6.22 mmol/L; hyper-LDL cholesterolemia, LDL-C $\geq$4.14 mmol/L; low-HDL cholesterolemia, HDL-C <1.04 mmol/L.

For laboratory evaluation, including the assessment of total TC, TG, HDL-C and LDL-C levels, five mL blood samples were collected from each individual. The blood samples were transported under refrigeration and than stored at −20 °C. Our study was approved by the ethics committee of the Jilin University School of Public Health (2012-R-011), and all subjects in this study signed a written informed consent form.

### SNP selection

Tag SNPs were selected from the Han Chinese data in Haploview (http://hapmap.ncbi.nlm.nih.gov/). Other functional or well-studied SNPs were simultaneously selected based on previous studies (*Wu et al., 2015*, *2016*) that have documented associations between SNPs in the *APOA1/C3/A4/A5-ZPR1-BUD13* gene cluster and lipid levels. The minor allele frequency (MAF) criterion for these SNPs was a MAF >0.05 in the Chinese Han population. Finally, six SNPs were included in this study:

*APOA1* rs5072, *APOC3* rs5128, *APOA4* rs5104, *APOA5* rs651821, *ZPR1* rs2075294 and *BUD13* rs10488698.

## DNA extraction and genotyping

Genomic DNA was extracted from peripheral blood lymphocytes following the protocols provided with a commercial DNA extraction kit (JiuNa Biology, Hangzhou, China). Genotyping of each SNP was performed via MALDI-TOF-MS (Sequenom, San Diego, CA, USA) with a MassARRAY system. Completed genotyping reactions were dispensed onto a 384-well SpectroCHIP using a MassARRAY nanodispenser (Sequenom, San Diego, CA, USA).

The detection rates for the six SNPs were 99.7%, 99.6%, 95.8%, 99.6%, 99.8% and 99.8% (for *APOA1* rs5072, *APOC3* rs5128, *APOA4* rs5104, *APOA5* rs651821, *ZPR1* rs2075294 and *BUD13* rs10488698, respectively).

## Statistical analysis

Continuous variables with a normal distribution were summarized as the mean ± standard deviation and compared by Student's *t*-test. The Chi-square test was used to check whether the genotype and allele distributions of the six SNPs were significantly different between males and females. For each SNP, the Hardy–Weinberg equilibrium (HWE) was calculated with the goodness-of-fit Chi-square test. $P < 0.05$ was considered statistically significant. Binary logistic regression analysis was used to detect associations between the genotypes of the six SNPs in the *APOA1/C3/A4/A5-ZPR1-BUD13* gene cluster (in the Dominant Model) and dyslipidemia adjusted for sex (adjusted only in the overall group), age, body mass index (BMI) and waist circumference in the overall, male and female groups. We set hypercholesterolemia, hypertriglyceridemia, high LDL cholesterolemia and reduced HDL cholesterolemia as dependent variables, while genotypes, sex (only in the overall group), age, BMI and waist circumference were set as independent variables and entered as method. The strength of any evident association was explored by calculating odds ratios (OR) together with their 95% confidence intervals (CI), and *P*-values of no more than 0.008 after Bonferroni Correction were considered statistically significant. All analyses were conducted using SPSS 24.0. Linkage disequilibrium (LD) and haplotype analyses were performed with the SNPStats program (https://www.snpstats.net/start.htm) (*Sole et al., 2006*) and Haploview software (*Barrett et al., 2005*).

## RESULTS

### Subject characteristics and distributions of genotype and allele

The baseline characteristics of the subjects stratified by genders were shown in Table 1. No difference was observed between males and females in terms of BMI, but the average age and waist circumference were greater in males than females. In addition to the percentage of subjects with hypercholesterolemia, there were significant differences in the percentages of subjects with hypertriglyceridemia, high LDL, reduced HDL and dyslipidemia between males and females. There were 2,264 participants without

**Table 1 Basic characteristics and distribution of dyslipidemia of included subjects.**

| Characteristics | Total ($n$ = 3,850) | Male ($n$ = 1,927) | Female ($n$ = 1,923) | $P$ |
|---|---|---|---|---|
| Age (year) | 49.54 (9.58) | 49.93 (9.75) | 49.16 (9.39) | **0.013** |
| Waist (cm) | 82.73 (11.22) | 84.74 (11.25) | 80.72 (10.83) | **<0.001** |
| BMI (kg/m$^2$) | 24.40 (3.96) | 24.38 (3.93) | 24.42 (4.00) | 0.734 |
| Hypertriglyceridemia, $n$ (%) | 1,110 (28.8) | 664 (34.5) | 446 (23.2) | **<0.001** |
| Hypercholesterolemia, $n$ (%) | 444 (11.5) | 214 (11.1) | 230 (12.0) | 0.406 |
| High LDL, $n$ (%) | 352 (9.1) | 145 (7.5) | 207 (10.8) | **<0.001** |
| Reduced HDL, $n$ (%) | 719 (18.7) | 450 (23.4) | 269 (14.0) | **<0.001** |
| Dyslipidemia, $n$ (%) | 1,586 (41.2) | 880 (45.7) | 706 (36.7) | **<0.001** |

Notes:
Means (standard deviation) for age, waist and BMI, a $P$-value in bold indicated differences were significant between males and females ($P < 0.05$).
BMI, body mass index.

dyslipidemia while the numbers of participants with one to four type of dyslipidemia were 763, 623, 184 and 16, respectively. The top five types of dyslipidemia (and the number of participants with each type) were as follows: hypertriglyceridemia (425), hypertriglyceridemia + reduced HDL cholesterolemia (397), reduced HDL cholesterolemia (220), hypercholesterolemia + high LDL cholesterolemia (126) and hypertriglyceridemia + hypercholesterolemia + high LDL cholesterolemia (107).

The genotype distributions of the six SNPs in the *APOA1/C3/A4/A5-ZPR1-BUD13* gene cluster conformed to HWE in the overall group (Table S1). The allele distribution of one SNP (rs651821) was different between males and females, and the genotype and allele distributions of the six SNPs were shown in Table S1. Genotype distributions and allele frequencies of dyslipidemia and normal groups were shown in Table S2.

## Association between SNPs and dyslipidemia

In the overall group, logistic regression analysis revealed that dyslipidemia was associated with three SNPs (rs5072, rs5104 and rs651821) after adjusting for sex, age, BMI and waist circumference (Table 2). In the male group, rs651821 was found to be associated with hypertriglyceridemia (OR = 1.78, 95% CI [1.41–2.26]) and low-HDL cholesterolemia (OR = 1.61, 95% CI [1.27–2.05]). Similar results were observed in the female group. However, we found that rs5072 was associated with hypertriglyceridemia (OR = 1.39, 95% CI [1.09–1.76]) in the female group (Table 2). The association between the numbers of risk alleles and dyslipidemia was shown in Table S3.

## Linkage disequilibrium and haplotype analyses

The pattern of pairwise LD between the SNPs of the *APOA1/C3/A4/A5-ZPR1-BUD13* gene cluster was shown in Fig. 1. Rs10488698, rs2075294 and rs651821 were located in block 1, while rs5104, rs5128 and rs5072 were located in block 2 after choosing the cut-off value of 0.75 for $D'$. The haplotype analysis results were shown in Table 3. People carrying haplotypes 2 and 3 presented an increased risk of dyslipidemia in the total sample

**Table 2 Association between genotypes of the six SNPs and dyslipidemia stratified by genders.**

| SNPs | Hypertriglyceridemia (n = 1,110) OR (95% CI)[a] | Hypercholesterolemia (n = 444) OR (95% CI) | High LDL (n = 352) OR (95% CI) | Reduced HDL (n = 719) OR (95% CI) |
|---|---|---|---|---|
| **Overall** | | | | |
| *APOA1*-rs5072 | | | | |
| TC + TT vs CC | **1.36 (1.15–1.60)** | 1.02 (0.83–1.25) | 0.84 (0.67–1.05) | 1.16 (0.97–1.39) |
| *APOA4*-rs5104 | | | | |
| GA + GG vs AA | 1.25 (1.06–1.48) | 1.01 (0.82–1.25) | 0.82 (0.65–1.03) | 1.24 (1.03–1.49) |
| *APOC3*-rs5128 | | | | |
| GC + CC vs GG | **1.33 (1.13–1.57)** | 1.02 (0.83–1.25) | 0.84 (0.67–1.05) | 1.16 (0.97–1.39) |
| *APOA5*-rs651821 | | | | |
| CT + CC vs TT | **2.01 (1.71–2.38)** | 1.23 (1.00–1.50) | 1.00 (0.80–1.25) | **1.90 (1.59–2.28)** |
| *ZPR1*-rs2075294 | | | | |
| GT + TT vs GG | 0.89 (0.74–1.07) | 0.83 (0.66–1.05) | 0.87 (0.67–1.13) | 0.91 (0.75–1.12) |
| *BUD13*-rs10488698 | | | | |
| CT + TT vs CC | 0.75 (0.59–0.95) | 1.02 (0.77–1.35) | 0.99 (0.73–1.36) | 0.70 (0.54–0.92) |
| **Males** | | | | |
| *APOA1*-rs5072 | | | | |
| TC + TT vs CC | 1.34 (1.06–1.69) | 1.16 (0.86–1.56) | 0.81 (0.58–1.15) | 1.19 (0.94–1.52) |
| *APOA4*-rs5104 | | | | |
| GA + GG vs AA | 1.28 (1.01–1.63) | 1.14 (0.84–1.54) | 0.77 (0.54–1.09) | 1.28 (1.00–1.63) |
| *APOC3*-rs5128 | | | | |
| GC + CC vs GG | 1.31 (1.04–1.65) | 1.20 (0.89–1.62) | 0.85 (0.60–1.20) | 1.16 (0.91–1.47) |
| *APOA5*-rs651821 | | | | |
| CT + CC vs TT | **1.78 (1.41–2.26)** | 1.16 (0.86–1.56) | 0.77 (0.55–1.10) | **1.61 (1.27–2.05)** |
| *ZPR1*-rs2075294 | | | | |
| GT + TT vs GG | 0.99 (0.76–1.30) | 0.68 (0.48–0.98) | 0.61 (0.39–0.95) | 1.03 (0.78–1.35) |
| *BUD13*-rs10488698 | | | | |
| CT + TT vs CC | 0.87 (0.62–1.20) | 1.24 (0.84–1.83) | 1.68 (1.10–2.57) | 0.74 (0.52–1.05) |
| **Females** | | | | |
| *APOA1*-rs5072 | | | | |
| TC + TT vs CC | **1.39 (1.09–1.76)** | 0.92 (0.69–1.22) | 0.87 (0.65–1.17) | 1.10 (0.84–1.45) |
| *APOA4*-rs5104 | | | | |
| GA + GG vs AA | 1.22 (0.96–1.55) | 0.91 (0.68–1.22) | 0.86 (0.64–1.17) | 1.18 (0.89–1.55) |
| *APOC3*-rs5128 | | | | |
| GC + CC vs GG | 1.37 (1.08–1.74) | 0.89 (0.67–1.18) | 0.85 (0.63–1.15) | 1.14 (0.87–1.50) |
| *APOA5*-rs651821 | | | | |
| CT + CC vs TT | **2.26 (1.77–2.88)** | 1.27 (0.95–1.69) | 1.19 (0.89–1.60) | **2.29 (1.73–3.04)** |
| *ZPR1*-rs2075294 | | | | |
| GT + TT vs GG | 0.82 (0.63–1.07) | 1.01 (0.73–1.38) | 1.11 (0.80–1.53) | 0.78 (0.57–1.07) |
| *BUD13*-rs10488698 | | | | |
| CT + TT vs CC | 0.66 (0.46–0.94) | 0.94 (0.62–1.42) | 0.64 (0.39–1.03) | 0.67 (0.44–1.02) |

**Notes:**
ORs (95% CI) were shown in bold if corresponding *P*-values were no more than 0.008 (0.05/6). The genetic model in this association analysis was the Dominant Model.

SNP, single nucleotide polymorphism; BMI, body mass index; OR, odds ratio; CI, confidence interval.

[a] ORs and 95% CI were adjusted for sex, age, waist circumference and BMI in the overall group; ORs and 95% CI were adjusted for age, waist circumference and BMI in the male and female groups.

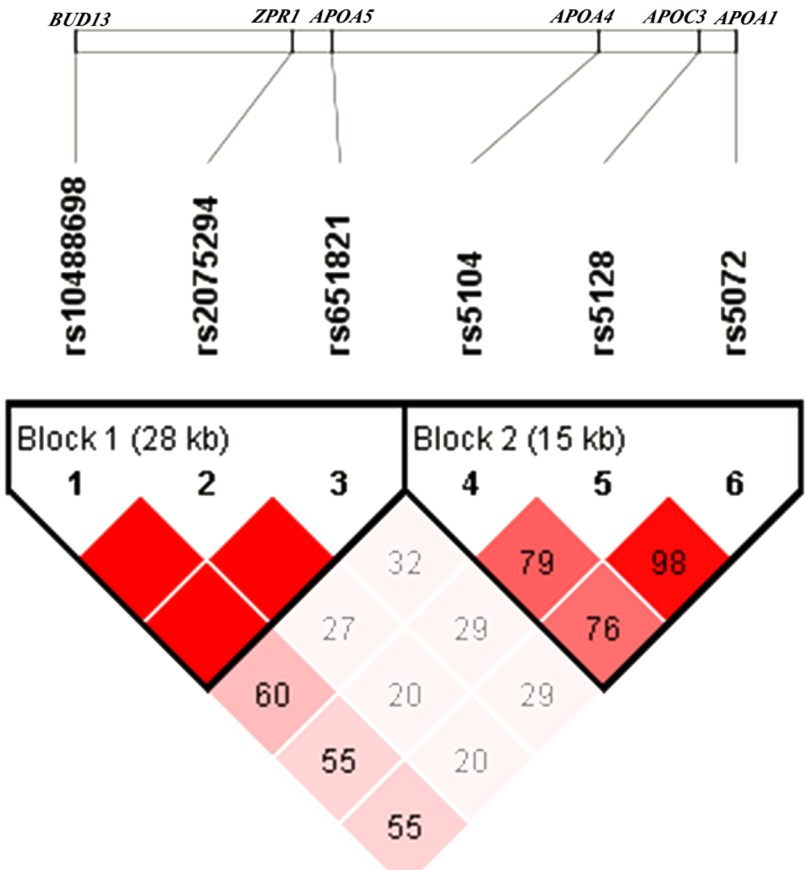

**Figure 1 Linkage disequilibrium analysis of six SNPs.** The value of $D'$ chosen as cut-off for linkage disequilibrium was 0.75.               

and the female and male groups. The association between haplotypes in blocks and different types of dyslipidemia stratified by sexes was explored. We found that subjects with haplotype CGC of block 1 exhibited a higher risk of developing hypertriglyceridemia and reduced HDL cholesterolemia in both sexes. In males, subjects carrying haplotype GCT of block 2 presented a higher risk of developing hypertriglyceridemia and reduced HDL cholesterolemia; however, these results were not observed in females. Details were shown in Table S4.

## DISCUSSION

The study applied a large sample design to study the association between the *APOA1/C3/A4/A5-ZPR1-BUD13* gene cluster gene polymorphisms and dyslipidemia. The participants in this study were enrolled from a representative community-based population. To the best of our knowledge, this is the first study to analyze the association of this gene cluster with dyslipidemia in a sex-specific pattern in northeast China.

The *APOA1/C3/A4/A5* gene cluster is a well-studied gene cluster. *Suárez-Sánchez et al. (2017)* found that there was a significant association between *APOA1* rs5072 and TG in Mexican children, and in an Indian study *Pranavchand & Reddy (2017)* concluded that

Table 3 Association between *APOA1/C3/A4/A5-ZPR1-BUD13* gene cluster haplotypes and the risk for dyslipidemia.

| Haplotype | SNPs[a] | | | | | | Number of variants | Frequency | Adjusted OR (95% CI)[b] | P[c] |
|---|---|---|---|---|---|---|---|---|---|---|
| | 1 | 2 | 3 | 4 | 5 | 6 | | | | |
| Overall | | | | | | | | | | |
| 1 | C | A | G | T | G | C | 0 | 0.3867 | 1.00 | – |
| 2 | T | G | C | C | G | C | 4 | 0.1297 | 1.63 (1.36–1.96) | **<0.0001** |
| 3 | C | A | G | C | G | C | 1 | 0.1145 | 1.65 (1.35–2.02) | **<0.0001** |
| 4 | C | A | G | T | T | C | 1 | 0.0673 | 1.02 (0.79–1.32) | 0.86 |
| 5 | C | A | G | T | G | T | 1 | 0.0644 | 0.83 (0.65–1.06) | 0.13 |
| 6 | T | G | C | T | T | C | 4 | 0.0615 | 1.05 (0.80–1.37) | 0.75 |
| 7 | T | G | C | T | G | C | 3 | 0.0614 | 0.92 (0.69–1.23) | 0.59 |
| 8 | T | A | C | T | G | C | 2 | 0.0251 | 0.61 (0.38–0.99) | 0.047 |
| 9 | C | G | G | C | G | C | 2 | 0.0187 | 1.43 (0.87–2.38) | 0.16 |
| 10 | C | G | G | T | G | C | 1 | 0.0153 | 0.87 (0.46–1.62) | 0.66 |
| 11 | T | A | C | C | G | C | 3 | 0.0119 | 2.35 (1.22–4.51) | 0.01 |
| 12 | C | G | G | T | T | C | 2 | 0.0119 | 1.54 (0.86–2.75) | 0.15 |
| Rare[d] | – | – | – | – | – | – | | 0.0316 | 1.20 (0.83–1.75) | 0.33 |
| Males | | | | | | | | | | |
| 1 | C | A | G | T | G | C | 0 | 0.3813 | 1.00 | – |
| 2 | T | G | C | C | G | C | 4 | 0.1396 | 1.43 (1.10–1.87) | **0.0078** |
| 3 | C | A | G | C | G | C | 1 | 0.1200 | 1.56 (1.16–2.10) | **0.003** |
| 4 | C | A | G | T | G | T | 1 | 0.0680 | 0.90 (0.64–1.28) | 0.57 |
| 5 | C | A | G | T | T | C | 1 | 0.0626 | 0.97 (0.65–1.44) | 0.88 |
| 6 | T | G | C | T | G | C | 3 | 0.0606 | 1.20 (0.79–1.81) | 0.39 |
| 7 | T | G | C | T | T | C | 4 | 0.0592 | 1.15 (0.77–1.73) | 0.50 |
| 8 | T | A | C | T | G | C | 2 | 0.0269 | 0.69 (0.36–1.32) | 0.26 |
| 9 | C | G | G | C | G | C | 2 | 0.0162 | 0.88 (0.36–2.11) | 0.77 |
| 10 | C | G | G | T | G | C | 1 | 0.0153 | 0.62 (0.25–1.53) | 0.30 |
| 11 | C | G | G | T | T | C | 2 | 0.0108 | 1.55 (0.61–3.95) | 0.36 |
| Rare[d] | – | – | – | – | – | – | | 0.0395 | 1.13 (0.69–1.88) | 0.62 |
| Females | | | | | | | | | | |
| 1 | C | A | G | T | G | C | 0 | 0.3932 | 1.00 | – |
| 2 | T | G | C | C | G | C | 4 | 0.1199 | 1.89 (1.45–2.45) | **<0.0001** |
| 3 | C | A | G | C | G | C | 1 | 0.1084 | 1.71 (1.29–2.28) | **2e-04** |
| 4 | C | A | G | T | T | C | 1 | 0.0720 | 1.12 (0.80–1.59) | 0.50 |
| 5 | T | G | C | T | T | C | 3 | 0.0638 | 1.04 (0.72–1.50) | 0.85 |
| 6 | T | G | C | T | G | C | 3 | 0.0620 | 0.68 (0.44–1.06) | 0.09 |
| 7 | C | A | G | T | G | T | 1 | 0.0602 | 0.80 (0.55–1.15) | 0.23 |
| 8 | T | A | C | T | G | C | 2 | 0.0237 | 0.67 (0.33–1.35) | 0.26 |
| 9 | C | G | G | C | G | C | 2 | 0.0216 | 1.94 (1.05–3.58) | 0.033 |
| 10 | C | G | G | T | G | C | 1 | 0.0149 | 1.39 (0.58–3.32) | 0.45 |
| 11 | T | A | C | C | G | C | 3 | 0.0144 | 2.76 (1.30–5.85) | 0.0081 |
| 12 | C | G | G | T | T | C | 2 | 0.0129 | 1.48 (0.69–3.16) | 0.31 |

| Haplotype | SNPs[a] | | | | | | Number of variants | Frequency | Adjusted OR (95% CI)[b] | P[c] |
|---|---|---|---|---|---|---|---|---|---|---|
| | 1 | 2 | 3 | 4 | 5 | 6 | | | | |
| 13 | T | G | C | T | G | T | 4 | 0.0115 | 1.52 (0.61–3.75) | 0.37 |
| Rare[d] | – | – | – | – | – | – | | 0.0214 | 1.16 (0.58–2.30) | 0.67 |

**Notes:**
SNP, single nucleotide polymorphism; BMI, body mass index; OR, odds ratio; CI, confidence interval.
[a] SNPs are as follows: 1, rs5072; 2, rs5104; 3, rs5128; 4, rs651821; 5, rs2075294; 6, rs10488698.
[b] ORs were adjusted for sex, age, BMI and waist circumference in the overall group and were adjusted for age, BMI and waist circumference in males and females.
[c] P-values > 0.008 were considered to be significant after Bonferroni correction and were presented in bold.
[d] Rare: haplotypes with frequencies <0.01.

rs5072 was also associated with increased TG; both of these findings were similar to our results. *Pranavchand & Reddy (2017)* found that the genotypic and allelic frequencies of rs5072 exhibited no significant differences between cases and controls; however, our study indicated that there was an association between them in females. One explanation for the difference between these two studies could be that they were conducted in people of different ethnicities and ages. A recent exome-wide association study (*Yamada et al., 2017*) performed in a Japanese population indicated that *APOA4* rs5104 located in this gene was associated with the serum TG concentration. A GWAS conducted in Chinese people (*Zhou et al., 2013*) showed that rs651821 presented a strong association with TG levels. *Cha et al. (2014)* drew the conclusion that subjects carrying the variant allele of this SNP exhibited a higher risk of developing low-HDL cholesterolemia than noncarriers in both genders in Koreans, which was consistent with our findings. In our study, haplotype analysis showed that participants carrying the CGC haplotype of block 1 presented a higher risk of developing hypertriglyceridemia (OR = 1.76, 95% CI [1.59–1.96]) and reduced HDL (OR = 1.80, 95% CI [1.59–2.03]) than those carrying the CGT haplotype, which revealed that the T allele of rs651821 was associated with a higher risk of hypertriglyceridemia and reduced HDL. The MAFs of rs5072, rs5128, rs5104 and rs651821 in our study were 0.3060, 0.2966, 0.2995 and 0.2406, respectively. These results were similar to the frequencies identified in the southern Han Chinese population in the 1,000 Genomes Project, which were 0.3190, 0.3190, 0.3571 and 0.2714. Compared with the frequencies observed in American subjects (0.0656, 0.0492, 0.1311 and 0.1639), British subjects (0.1099, 0.1209, 0.1648 and 0.0714) and other ethnicities, the frequencies determined in our study were markedly different. Ethnicity may be critically important in studying the association between SNPs and dyslipidemia. We used Polyphen to predict the influence of two missense mutations (*APOA4* rs5014 and *BUD13* rs10188698) on the resultant proteins and found that there was no significant influence on protein function. We also observed that these two SNPs were not associated with any types of dyslipidemia.

At present, studies on *BUD13-ZPR1* gene polymorphisms focus on the association with lipid-related diseases. *BUD13* and *ZPR1* were found to be associated across anatomical categories of coronary artery disease in an Indian study (*Pranavchand, Kumar & Reddy, 2017*). Another study (*Lin et al., 2016*) performed in a Taiwanese population

indicated that *BUD13* may contribute to the risk of MetS, and *BUD13* rs623908 was shown to be significantly associated with high TG, low HDL and HDL levels. Among Chinese individuals (*Xu et al., 2018*), a strong association between *BUD13-ZPR1* rs964184 and coronary heart disease was found, and both gender and age had great impacts on the association of the rs964184 polymorphism with coronary heart disease. Another study (*Ueyama et al., 2015*) indicated that rs964184 was significantly associated with the prevalence of MetS. In addition, there have been studies showing that that *ZPR1* contributed to the risk of type 2 diabetes mellitus (*Guan et al., 2016*; *Tokoro et al., 2015*). Due to the association of these two genes with lipid-related diseases, we conjecture that *BUD13* and *ZPR1* gene polymorphisms may contribute to the risk of dyslipidemia.

In our study, we obtained the novel finding of sex-specific differences between the *APOA1/C3/A4/A5-ZPR1-BUD13* gene cluster gene polymorphisms and dyslipidemia. These differences could be partially explained by differences in sex between males and females. Menopause could change the lipid profile by reducing HDL-C and increasing TG, TC and LDL-C (*Reddy Kilim & Chandala, 2013*). However, after menopause, healthy females were prone to become more insulin resistant and gain total body and intra-abdominal fat (*Milewicz, Tworowska & Demissie, 2001*). Additionally, females with polycystic ovary syndrome usually become overweight or obese, especially abdominally obese (*Wu & Von Eckardstein, 2003*). Accumulation of body and central fat and decreased insulin resistance are associated with TG, HDL-C and LDL-C and may therefore alter lipid homeostasis, respectively (*Ferrannini et al., 2007*). Consequently, mechanisms underlying sex-specific differences between the gene cluster and dyslipidemia are complex and should be further studied. We also found that the associations of the SNPs with dyslipidemia were different when the total sample, males and females were considered separately. We speculated that these associations may be influenced by sex and $P$-values we set, since some $P$-values were between 0.008 and 0.05. There are some limitations of our study that must be recognized. In each gene, we selected only one tag SNP with a MAF >5% reported in published articles. We may have ignored some SNPs with low MAFs or important functions in the *APOA1/C3/A4/A5-ZPR1-BUD13* gene cluster. In addition, dyslipidemia is influenced not only by genetic factors but also by diet, exercise and other environmental factors. We adjusted only sex, age, BMI and waist circumference in our study. Consequently, more environmental factors need to be recorded and adjusted in future studies, and the effects and mechanisms of SNPs in relation to gene expression and protein expression warrant further study.

## CONCLUSION

In conclusion, we analyzed and assessed the association between *APOA1/C3/A4/A5-ZPR1-BUD13* gene cluster gene polymorphisms and dyslipidemia between sexes. We not only obtained several results consistent with those reported based on previous studies but also found one SNP (rs5072) was different between males and females, providing new insight into the mechanisms of dyslipidemia in different genders.

### Funding

This study was supported by the Scientific Research Foundation of Jilin Provincial Health Department, China (#2011Z116). The funders had no role in study design, data collection and analysis, decision to publish, or preparation of the manuscript.

### Grant Disclosure

The following grant information was disclosed by the authors:
Scientific Research Foundation of Jilin Provincial Health Department, China: #2011Z116.

### Competing Interests

The authors declare that they have no competing interests.

### Author Contributions

- Wei Bai conceived and designed the experiments, performed the experiments, analyzed the data, contributed reagents/materials/analysis tools, prepared figures and/or tables, authored or reviewed drafts of the paper, approved the final draft.
- Changgui Kou conceived and designed the experiments, performed the experiments, analyzed the data, authored or reviewed drafts of the paper, approved the final draft.
- Lili Zhang performed the experiments, analyzed the data, authored or reviewed drafts of the paper.
- Yueyue You performed the experiments, analyzed the data.
- Weiying Yu contributed reagents/materials/analysis tools, prepared figures and/or tables.
- Wanqing Hua contributed reagents/materials/analysis tools, prepared figures and/or tables.
- Yuanyuan Li contributed reagents/materials/analysis tools, prepared figures and/or tables.
- Yaqin Yu conceived and designed the experiments.
- Tiancheng Zhao conceived and designed the experiments, performed the experiments, authored or reviewed drafts of the paper, approved the final draft.
- Yanhua Wu conceived and designed the experiments, performed the experiments, authored or reviewed drafts of the paper, approved the final draft.

### Human Ethics

The following information was supplied relating to ethical approvals (i.e., approving body and any reference numbers):

The Ethics Committee of Jilin University School of Public Health granted Ethical approval to carry out the study (2012-R-011).

### Data Availability

The raw data for dyslipidemia analysis are provided in the Supplemental Files.

## Supplemental Information

Supplemental information for this article can be found online at http://dx.doi.org/10.7717/peerj.6175#supplemental-information.

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
