# Peer review of "Functional polymorphisms of the APOA1/C3/A4/A5-ZPR1-BUD13 gene cluster are associated with dyslipidemia in a sex-specific pattern"

_PeerJ, doi:10.7717/peerj.6175_

## Round 0.1 · original submission · Major Revisions

Dear authors,

Your manuscript has been carefully reviewed by three experts in the topic and all of them have indicated your paper has high scientific standards to be considered for publication in PeerJ. However, they have highlighted some modifications which you should take into acount in a revised version of your manuscript (MAJOR REVISION).

With respect and warm regards,
Dr Palazón-Bru (academic editor for PeerJ)

Reviewer 1 ·

Basic reporting

Please see attached PDF.

Experimental design

Please see attached PDF.

Validity of the findings

Please see attached PDF.

Additional comments

Please see attached PDF.

Annotated reviews are not available for download in order to protect the identity of reviewers who chose to remain anonymous.

Reviewer 2 ·

Basic reporting

The present article aimed to investigate the association between six polymorphisms located in APOA1/C3/A4/A5-ZPR1/BUD13 gene cluster and dyslipidemia in a sex-specific pattern. I will list some important topics that should be taken into consideration by the authors and be better addressed before publication.
Introduction/Background Section has recent references from literature contextualizing the importance for the studied theme. However, the authors failed in explaining why they have chosen these specific genes and/or polymorphisms for further investigation. Lines 216-227 and 247-266 (Discussion Section) should be better replaced within the Introduction Section because these paragraphs could better explain the function of the selected genes and how they influence lipid levels contributing to the risk of dyslipidemia.
In line 112 (SNP Selection Section), the authors mentioned that these SNPs were selected according to minor allele frequency and based on previous studies; however, they did not mention the allele frequencies observed for these polymorphisms in the literature and did not provided references for this sentence.
The English language is adequate and clear in most of the text, but the authors should be more objective and concise mainly in the Results Section. Sometimes, it is difficult to follow the description of the results because of the amount of detailed information that was presented. The authors should try to reduce both the text size and the number of Tables by focusing in the presentation of the most relevant results. It would be easier to understand if the authors joined both the results of Chi-Square tests (Supplementary Tables 1 and 2) and logistic regression models with adjusted OR values (Table 2) in one single Table.
Table 1 should present the names of the genes together with the polymorphisms. It would be also interesting to have a complete description of other diseases and/or serum levels collected from the same individuals.

Experimental design

The authors focused in showing the results separately by gender, but it should be interesting to present the results in the total sample before stratifying it in males and females groups. They could be missing important information that is not gender-related.
The sample size is big (N=3850) and individuals were randomly selected from another study containing 21435 individuals. The authors did not explain if the sample size was previously calculated considering the frequencies of the studied SNPs.
The studied population is part of a project that aimed to evaluate the prevalence and risk factors associated with chronic diseases (as mentioned in line 93 – Materials and Methods Section). Therefore, the authors probably have access to more detailed information regarding the patients included in this study (e.g. other co-morbidities or serum levels). They should also provide the genetic association results for these other studied variables (mainly those related to dyslipidemia, such as cardiovascular disease, obesity, stroke or other metabolic-related diseases).
I would suggest for the authors to analyze the Hardy-Weinberg Equilibrium (HWE) both in the entire sample and separately by gender. Also, the results of HWE should be presented as Supplementary Table.

Validity of the findings

Regarding the Statistical Analysis Section, lines 127 to 129 described that the authors chose statistical tests considering distributions that followed or not the normality of the continuous variables. However, once they have previously transformed the numeric variables to logarithmic equivalents, they could use only parametric tests. Furthermore, lines 132-133 and lines 136-138 described the same analyses (Chi Square test + Logistic Regression Models with OR and 95% CI adjusted for covariates) and both sentences could be joined in one single sentence. Moreover, once there are different polymorphisms under investigation as well as several independent tests being performed for each SNP, the authors should have performed a correction for multiple testing (e.g. Bonferroni Correction).
The authors showed the association of the polymorphisms with Dyslipidemia (as categorical variables), but also the association results with lipid levels as numeric variables. However, Dyslipidemia was characterized considering the altered lipids levels and, in my opinion, it is redundant to present both results. Therefore, it would be better if the authors could focus their analyses by choosing only one type of variable (categorical or numeric). For example, it was confusing when comparing the results presented in Tables 2 and 3, not only because of the lack of concordance between the presented results, but also because they were not performed using the same genetic models.
The authors should better explain the genetic models that are under investigation for the association analysis. It was not clear why they are presenting different genetic models for different analyses. They should choose the best genetic models and use the same ones for all the analyses. For example, Supplementary Tables 1 and 2 showed the results of Chi-Square test associating polymorphisms and dyslipidemia in males and females, respectively, using Genotype and Allelic Models. However, Table 2 showed the adjusted OR values (and 95% CI) for dyslipidemia considering Dominant and Allelic Model. Also, while Table 2 presented the results using the Dominant and Allelic Models, Table 3 used only the Genotype Model.
The authors only performed association analysis considering the individual polymorphisms. Once they are located within the same gene cluster, they should provide the distance among them and also perform linkage disequilibrium and haplotype association analyses.

Additional comments

There are several points that the authors should address to make the manuscript acceptable for publication. I pointed some concerns and limitations and gave some suggestions for the improvement of the presented article.

Reviewer 3 ·

Basic reporting

Some sections in your manuscript need more detail and my suggestions can be found below.
- In line 66 it is stated that “genetic factors could also partially explain the development of dyslipidemia”. This has been already proven, probably not in Chinese population, but the genetic control of lipid parameters is well-documented (for example by Weiss et al (Nat Genet. 2006;38:218) or Dron et al (Curr Genet Med Rep. 2016;4:130 and Curr Opin Lipidol. 2018;29:133). Therefore, the statement should include this information.
- In the Background section we can find the description of the gene cluster APOA1/C3/A4/A5-ZPR1-BUD13 (from line 68 to 86) particularly involved in lipid metabolism. In the paragraph the cluster is written in different ways (in upper- and lower-case letters, with more or less components) and it should be uniformed. Upper-case letters are commonly used when referring to a gene and lower-case are used when referring to proteins.
- The first time a contraction is used, it should be described. Despite that, the description of a gene name may also help understanding its role, location, etc. For example, ZPR1 or BUD13 are used in the Background section and should be described. The APO genes cluster is very well-known but the other two genes are less recognized.
- The Figures should include the sample size per group in order to improve its self-explanation.

Experimental design

In the Methods Section there is a description of the study population, however some useful information regarding the participants is missing.
- when describing the study population it is not clearly stated whether the participants are being treated with any drugs that may affect lipid parameters;
- it would be very informative if the sample size for each group of dyslipidemia was showed there;
- it would be very interesting knowing whether there are any participants in more than one dyslipidemia group (for example presenting with increased TG and lowered HDL, or with high TG and high LDL). How many participants do not have any altered lipidic parameter?

Validity of the findings

Since the genetic control of lipid parameters is very well-known for being complex, and since the authors stated in the Background section that different polymorphisms in the cluster have been already described for being associated with dyslipidemia, is there any participant carrying more than one of the studied genetic variants? Is there an allele-dose effect meaning that those participants with more than one rare allele present with a worse lipid profile than those with 1 or none?

---

## Round 0.2 · Major Revisions

Dear authors,

Your manuscript still needs some modifications in a revised version which you should submit to PeerJ in order to consider again your work. Please, see the comments of the reviewers in order to have more information.

With respect and warm regards,
Dr Palazón-Bru (academic editor for PeerJ)

Reviewer 2 ·

Basic reporting

In the second paragraph of the Background Section, the authors cannot say that the polymorphisms are influencing the transcription factors and/or miRNA binding to their corresponding sequences unless they provide an in silico prediction for this inference. The same occurs when they say that the studied polymorphisms located in coding regions could lead directly to missense mutations and influence gene expression. There are available softwares (e.g. SIFT and Polyphen) that allow predicting if a missense alteration is pathogenic or tolerant for the protein function.
The authors should be careful with the gene names, because they must be written always in capital letter and in italic (please, correct both text and tables). Also, in the third paragraph of the Background Section, authors should use capital letters to describe the human protein names.
There are missing references for the last paragraph of the Background Section.

Experimental design

Table S1 describes the genotype and allele distributions of the six SNPs in the overall sample and separated by sex. The authors did not specify if the p-value is for the comparison of the distribution between males and females. Furthermore, there is no statistical description for this analysis. Also, the authors did not mention anything regarding the significant p-value (p=0.032) found for the rs651821 polymorphism. Is it really significant? The distribution of the frequencies between genders did not seem to be different.
Hardy Weinberg Equilibrium (HWE) should not be calculated only considering the individuals without dyslipidemia. The authors should consider the total sample (independently of the dyslipidemia) for calculating the HWE. Also, the authors must correct for “Hardy Weinberg Equilibrium” instead of “Hardy-Weinberg disequilibrium”. HWE could be placed in the same table as the genotype and allele distributions (Table S1), in an additional column.
Regarding haplotype analysis, Table 3 showed the results of the associations between dyslipidemia and haplotype blocks including all the studied polymorphisms, but the authors did not analyze if all of them were in linkage disequilibrium. In Table S4, the authors provided only the linkage disequilibrium results (D’ values) for pairs of polymorphisms, separately.
The authors only showed the association regarding the haplotype blocks and dyslipidemia. Why did they not provide association between haplotype blocks and Hypertriglyceridemia, Hypercholesterolemia, high LDL and reduced HDL, separately?
I strongly recommend the use of Haploview software for linkage disequilibrium and haplotype association analyses. The authors should mention the value of D’ chosen as cut-off for linkage disequilibrium analysis and also the distance among the polymorphisms in the haplotype blocks.

Validity of the findings

In the previous version, I have recommended for the authors to join Table S1, Table S2 and Table 2, but the authors decided to delete the Chi-Square results from the revised manuscript. It would be interesting to keep the results of the Chi-square test showing the associations between dyslipidemia and the polymorphisms (in the Genetic Model chosen for this analysis) and, in the same Table, add the adjusted OR values which were calculated by the binary logistic regression.
In Table 2, the authors could present only the adjusted Odds Ratio values. The footnotes should describe the covariates which were chosen for each analysis. They did not discuss the reasons or hypotheses why the associations of the SNPs with dyslipidemia are different when considering the total sample or males and females, separately.
In my opinion, the authors should investigate if the dyslipidemia-related diseases needed to be included as covariates for adjusting the OR values in the logistic model (even if this association have been previously described in a previous article). In the same way, do the authors have access to information regarding the use of medications? All the mentioned factors can clearly modify association between polymorphisms and metabolic variables and dyslipidemia.
Haplotype analysis was poorly described and discussed.

Additional comments

Once the authors performed other analyses as recommended by Reviewers, other concerns and limitations appeared. There are several genetic conceptual mistakes that should be taken into consideration and improved by the author. E.g. correct the gene names, Hardy Weinberg Equilibrium (instead disequilibrium), correct the calculation of the HWE, analyze other important covariates (e.g. medications and dyslipidemia-related diseases) that should be included for adjustment in the binary logistic model, improvement of the linkage disequilibrium and haplotype analyses, etc

Reviewer 3 ·

Basic reporting

The Background section has been modified following reviewer’s suggestions. Information regarding the genes in the apo cluster, which was previously detailed in the Discussion, has been moved to the Background section. However, now there is a clear lack of information describing the other two genes (ZPR1 and BUD13).

Experimental design

Some aspects have been corrected in the Methods section following reviewer’s suggestions. However, the paragraph describing the logistic regression analysis to detect associations with dyslipidemia, now is not clear enough.
In the Results section, the authors added information describing the amount of participants presenting with one or more dylipidemic components, following my suggestion. However the data is not clear enough presented. The authors stated “participants with one to four type of dyslipidemia were 763, 623, 184 and 16, respectively” but it would be very interesting knowing which types of dyslipidemia are shared when presenting either two, three or four.

Validity of the findings

The authors have added some data regarding a dose-allele effect, and apparently presenting with more than one genetic variant is associated with increased risk of dyslipidemia. In fact, in their cohort, a high percentage of participants (84.2%) present with more than 1 studied genetic variants. It would be very informative if the authors could add a table describing the different combinations of genetic variants found in their cohort, and the number of participants sharing each combination.

---

## Round 0.3 · Minor Revisions

Dear authors,

Still pending some minor changes before accepting your paper for publication in PeerJ.

With respect and warm regards,
Dr Palazón-Bru (academic editor for PeerJ)

Reviewer 2 ·

Basic reporting

The authors presented the association results in total sample and also stratified by genders, as suggested before. Therefore, they should include this aim in the Abstract and also in the last paragraph of the Introduction Section (lines 106 -111: “This study assessed the association of tag and well-studied SNPs … with dyslipidemia in overall sample and stratified by gender in a large sample from a Han Chinese population”).
The authors performed haplotype association analysis, but there is no information regarding this topic (both description and results) in the Abstract Section.
The authors forgot to correct the gene cluster name in line 127 of the revised manuscript (Materials and Methods Section). The gene names should be written in italic. Also, in the third paragraph of the Background Section, authors should write the entire name of the human protein names in capital letters, and not only the first letter.
Text and English language should be carefully reviewed because there are several mistakes throughout the text (e.g. words written together; the use of singular instead of plural; sentences with lack of verb; please, correct the word PolyPhen (line 253 of the revised manuscript), etc).
I will point some other revisions to be correct in tables and figures:
- In the footnote of Table 1, please correct “Means (standard deviation)” instead of “Means ± standard deviation”. Please, include the abbreviation meaning of BMI in the footnote.
- In the title of Table 2, please correct “Association between genotypeS of the 6 SNPs and dyslipidemia in overall sample and stratified by genderS”. Furthermore, the word “LDL” is missing in the fourth column (“HIGH LDL”). Please, include the abbreviations meaning (SNP, BMI, OR and CI) in the footnote.
In the title of Table 3, correct “risk FOR dySlipidemia”. In the footnote, please explain the reason for p<0.008 (for example: “p<0.008 were considered as significant after Bonferroni Correction and were represented in bold”). Please, include the abbreviations meaning (SNP, BMI, OR and CI) in the footnote.
In Figure 1, the authors should include the order of the genes analyzed in the haplotype blocks to make it easier for the reader.
In Table S1, significant results (p<0.05) should be highlighted in bold and also be mentioned in the footnote.
In Table S2, please correct the word NORMAL (instead of nomal), in the third column. In Table S3, please include the abbreviations meaning (BMI, OR, CI) in the footnote and also highlight the significant results. Please, correct “number of risk alleleS”, using plural instead of singular in both title and table.

Experimental design

Regarding haplotype analysis, why did the authors not perform haplotype association analysis stratified by genders? In Figure 1, the authors could be less strict when choosing the cut-off value for D’. They chose D’>0.90 as cut off for considering the polymorphisms in linkage disequilibrium and, therefore, they could be losing important associations. If they choose D’> 0.75, rs5104 could be included in the haplotype block 2.
In Table S2, why did the authors present only the association results stratified for genders? They should also present the results found in the overall sample (as performed in Table 2).

Validity of the findings

The authors should compare the frequencies they found with other populations in the literature (for example with the frequencies described in 1000 Genome Project).

Additional comments

The authors have revised the manuscript and answered the suggestions that the reviewers have made. However, I will suggest other revisions to make it suitable for publication in Peer J.
The main concern for me at this point is the correction of the language and the presence of several errors found throughout the text. I will point some of them, but the authors should carefully review the manuscript before publication.

Reviewer 3 ·

Basic reporting

Due to the changes in the Results section, the English should be revised throughout. For example, the verb tenses used when referring to Tables/Figures should be consistent; and some sentences are not clear or easy-to-understand.

Experimental design

no comment

Validity of the findings

no comment

---

## Round 0.4 · accepted · Accept

Dear authors,

I am pleased to inform that your paper has been accepted for publication in PeerJ.

Congratulations!

With respect and kind regards,
Dr Palazón-Bru (academic editor for PeerJ)


# Reviewer 2 ·

Basic reporting

The authors should correct the footnotes of both Table S1 and S2 “p values less than 0.05 were presented in bold” in place of “no more than 0.05”. Also, the authors should include a legend for the Figure 1, indicating the meaning of the color and the numbers inside the squares of the figure.

Experimental design

No comments

Validity of the findings

No comments

Additional comments

The authors clearly answered all the pointed questions, corrected the mistakes and improved the language. In my opinion, it is now suitable for publication in Peer J.

Reviewer 3 ·

Basic reporting

no comment

Experimental design

no comment

Validity of the findings

no comment

Additional comments

The authors have revised the manuscript answering the comments and suggestions made by the reviewers. In my opinion the manuscript has improved and is now ready for publication, therefore I have no additional comments.